# Conditional Diffusion Model for Open-ended Video Question Answering

## ABSTRACT

Open-ended VideoQA presents a significant challenge due to the absence of fixed options, requiring the identification of the correct answer from a vast pool of candidate answers. Previous approaches typically utilize classifier or similarity comparison on fusion feature to yield prediction directly, lacking coarse-to-fine filtering on numerous candidates. Gradual refining the probability distribution of candidates can achieve more precise prediction. Thus, we propose the DiffAns model, which integrates the diffusion model to handle open-ended VideoQA task, simulating the gradual process by which humans answer open-ended question. Specifically, we first diffuse the true answer label into a random distribution (forward process). And under the guidance of answer-aware condition generated from video and question, the model iteratively denoises to obtain the correct probability distribution (backward process). This equips the model with the capability to progressively refine the random probability distribution of candidates, ultimately predicting the correct answer. We conduct experiments on three challenging open-ended VideoQA datasets, surpassing existing SoTA methods. Extensive experiments further explore and analyse the impact of each modules, as well as the design of diffusion model, demonstrating the effectiveness of DiffAns. Our code will be available.

## CCS CONCEPTS

• Information systems → Question answering; Multimedia and multimodal retrieval.

## KEYWORDS

Video Question Answering, Video-Language, Diffusion Model

## 1 INTRODUCTION

Video Question Answering (VideoQA) seek to explore technique that combines video when answering question [50], representing one of the challenging multimodal tasks. The VideoQA task can be categorized into two types: multi-choice (several candidate answers are provided for each question) [23, 44] and open-ended (a very large candidate answers set are constructed for all questions)[18, 48, 49]. In this paper, we focus on the open-ended VideoQA which is more challenging than multi-choice VideoQA.

**Unpublished working draft. Not for distribution.**

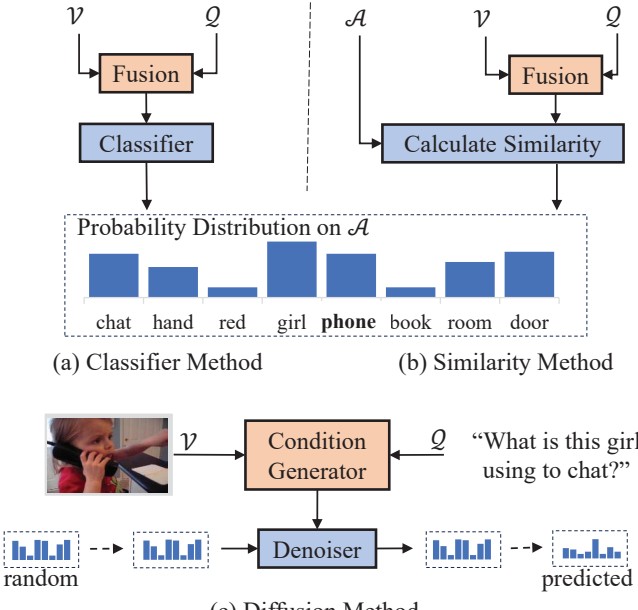

(a) Classifier Method      (b) Similarity Method

(c) Diffusion Method

**Figure 1: Comparison between our diffusion method and other methods.**

Open-ended VideoQA is popularly set as a multi-class classification problem, where the means of answer generation are mainly divided into two categories: classifier and similarity method. Classifier method directly use a classifier to predict the answer [12, 27, 45, 51], as illustrated in Figure 1(a). Initially, the features of video and question are modeled for interaction and fusion, followed by the usage of a classifier to classify the fused feature on a large candidate answer set. Cross-entropy is employed as the loss function during training, while the candidate corresponding to the highest probability score is selected as the answer during prediction. This method is straightforward but may lose the textual semantic information of answer while treating candidates solely as classification labels. Therefore, some recent methods [5, 46, 47] attempt to encode the textual content of candidates as features, aiming to optimize the answer prediction semantically, as depicted in Figure 1(b). During training, they employ contrastive learning to minimize the distance between the fused features of video-question pairs and the features of ground truth answers. Then they select answer with the highest similarity to the fused features as the prediction.

These two methods, while both achieving decent results, select only one answer from a vast candidate set directly without coars-to-fine filtering, inevitably leading to significant inaccuracy in answer prediction. According to the general intuition when facing open-ended question, human typically continuously search for

the direction, narrow down the scope of candidates and finally respond more precisely. For example, given the video and question in Figure 1(c), we can first come up with some more relevant answers (with a higher probability), and then lock in the correct answers ("phone"). This characteristic is what the model precisely needs when handling open-ended VideoQA. Excitingly, the recent diffusion models[33, 38], which are popular and applied in various fields [6, 20, 25, 43], precisely align with this advantage. Their notable effectiveness lies in progressively generating vivid images from noisy images, approaching real scenes step by step. Thus, we incorporate the gradual refining from random to precise features of diffusion models to simulate human intuition for open-ended VideoQA, as shown in Figure 1(c). Firstly, we interact with the video and question to generate condition aware of answer. Then, we introduce a random probability distribution of candidate answers and use a denoising network based on diffusion models to incrementally refine the probability distribution of answers, guided by the condition possessing fusion information of video and question. In this way, the model continuously iterates to search for the direction and narrow down the scope of answers until finally locking onto the answer accurately. This achieves a more precise process of selecting the unique correct answer from a vast candidate set.

Therefore, to model the open-ended VideoQA process mentioned above, we propose DiffAns to tackle open-ended VideoQA with Diffusion Model. Specifically, we first embed the video and question using pre-trained image and text backbones, and achieve semantic projection of the visual modal and contextual perception of question. Next, we construct the Answer-aware Condition Generator for generating condition aware of answre for the subsequent diffusion model. It handles the video and question features through interaction and fusion, extracting the key information required for answering which ensures the correct distribution updating during denoising. After that, the proposed Answer Denoiser encode the answer probability distribution and denoises it with noise intensity guided by answer-aware condition. Then, it decodes the denoised feature yielding the distribution at next step. Iterate that process and finally obtain the correct answer probability distribution.

Our contributions can be summarized as follows:

- We tackle the open-ended VideoQA task with diffusion model and propose DiffAns. It emulates human intuition by iterative seeking answer direction and narrowing down scope of the answer to select the unique correct answer from a large candidate set. The coarse-to-fine refining process achieving a more precise answer prediction.
- We design the Answer Denoiser to enhance the denoising process of diffusion model. It entails encoding and decoding answer probability distribution and incorporates noise intensity to refine the noisy distribution under the guidance of answer-aware condition.
- We conduct comparative experiments on three commonly used and challenging open-ended VideoQA datasets, demonstrating the superiority of our method over existing SoTA approaches. Additionally, our extensive experiments further explore and analyse the impact of each modules, as well as the design of diffusion model.

## 2 RELATED WORK

### 2.1 Open-ended VideoQA

Open-ended VideoQA is aimed at comprehending video content to answer to related questions without provided answer options, unlike multi-choice VideoQA[50]. Thus, a vast answer set is often constructed as global answer options specific to the dataset. Consequently, each answer is treated as class label and the task is transformed into a multi-label classification task[50]. In recent years, two predominant solutions have emerged: classifier method and similarity method. Classifier method is more prevalent, involving various forms of interaction modeling between video and question, which can be spatio-temporal[7, 11, 19, 21], hierarchical[7, 22, 29, 34, 45], multi-scale[12, 34], multi-granularity[45], or Transformer-based[27, 28, 51]. And the final fusion of feature is fed into a classifier. This method typically employ cross-entropy loss for model training, and select the label with the highest probability score as the predicted answer. However, such straightforward approaches to a certain extent sacrifice the intrinsic semantic information of each answer text. Recently, some endeavors [5, 46, 47] have sought to harness the semantic of answers to guide the training and inference process of model. During the training phase, they employ contrastive learning as a loss function to steer the distribution of fusion features and answer features after interacting and fusing the video and the question. This method treats the label answer as positive sample while considering other answers as negative samples, aiming to align the distributions of fused feature with the label answer closer and diverge from other candidate answers. In the inference phase, they compute the dot product similarity between the fused feature and all candidate answers, selecting the answer with the highest similarity as the prediction. Despite the respectable performance achieved by both methods, the process of directly selecting the sole answer from an extensive answer set in one step raises doubts. We aspire for models to progressively refine and determine answers step by step to possess the coarse-to-fine filtering of singular answer selection. Thus, we leverage the corresponding properties of diffusion models to tackle open-ended VideoQA. We realize an intuitive approach, iteratively narrowing down the range of answers and ultimately predicting the correct answer. We use denoiser of diffusion model to mimick the process of iteratively refining answers from the vague recesses of the mind in accordance with human intuition.

### 2.2 Diffusion Models

The diffusion model [17, 33, 41] belongs to generative models, which simulate diffusion processes in physical thermodynamics. During the training process, it initially adds noise to the labels through a forward process until they approximately conform to a Gaussian distribution. Subsequently, in the reverse process, it learns to reverse this noisy process through a denoiser to obtain the original labels. In recent years, the remarkable success of this model in the generative domain has been exhilarating, especially in the field of visual generation[2, 38, 40], which have generated high-quality images that are difficult to distinguish between real and fake, completely overturning previous generative models. Recently, in the domain of video generation, models [3, 16] have similarly produced stunning generated videos using diffusion models. Due

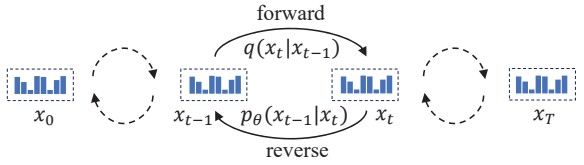

**Figure 2: The forward and reverse process of diffusion model.**

to its progressive refinement of labels and the continuous improvement in training and inference efficiency, it has also been applied in various fields such as Image Detection[6], recommend system[43], Video-Text Retrieval[20], and Video-Text Localization[25]. In the early stages, the denoising networks of diffusion models were typically dominated by U-Net[39], but recently, Transformer [42] have been gradually explored and applied, achieving remarkable results. However, to our knowledge, researchers have not yet utilized diffusion models to tackle VideoQA. Therefore, to address this gap, we fully leverage the characteristics of diffusion models to handle open-ended VideoQA in a more precise manner. Meanwhile, we implement the denoiser of the diffusion model using transformer-based modules. To the best of our knowledge, we are the first to adapt the diffusion model for VideoQA.

## 3 PRELIMILARIES

To better comprehend our utilization of the diffusion model in achieving a coars-to-fine answer process for open-ended VideoQA, we first describe the definition of open-ended VideoQA, and then clarify the theory of the forward and reverse processes within the diffusion model.

### 3.1 Open-ended VideoQA Definition

For open-ended VideoQA, the input typically comprises an untrimmed video (i.e., $\mathcal{V} = \{v_i\}_{i=1}^{N_v}$) and a corresponding question (i.e., $Q = \{q_i\}_{i=1}^{N_q}$), where $N_v$ and $N_q$ denote the number of frames and words respectively. Concurrently, there exists an extensive answer set (i.e., $\mathcal{A} = \{a_i\}_{i=1}^{N_a}$) aiming to encompass all possible answers specific to dataset, where $N_a$ represent the number of candidate answers. The model necessitates modeling the function $\Omega$ process to achieve the selection of the correct answer $a^* = \Omega_\theta(\mathcal{V}, Q)$, where $a^* \in \mathcal{A}$ and $\theta$ denotes the parameters of model.

### 3.2 Forward and Reverse Process

Forward process is essentially a noisy process that requires continuously adding noise to the original label information $x_0 \sim q(x_0)$ to obtain noisy data $x_T$ after $T$ steps, where $T$ represents the intensity of the noise. As depicted in Figure 2, in each step, the forward process of adding noise [17] can be represented as

$$q(x_t|x_{t-1}) = \mathcal{N}(x_t; \sqrt{1-\beta_t}x_{t-1}, \beta_t I), \quad (1)$$

where $\beta$ denotes the variance schedule. The process to get $x_T$ in $T$ steps can be defined as

$$q(x_{1:T}|x_0) = \prod_{i=1}^{T} q(x_t|x_{t-1}). \quad (2)$$

Through reparameterization technique, the final noisy distribution $x_T$ can be directly represented as

$$x_T = \sqrt{\bar{\alpha}_T}x_0 + \sqrt{1-\bar{\alpha}_T}\epsilon_T, \quad (3)$$

where the $\epsilon_T \sim \mathcal{N}(0, I)$ denotes the noise and $\bar{\alpha}_T = \prod_{i=1}^{T}(1-\beta_i)$.

Reverse process is a denoising process, which continuously denoises the noised distribution $x_T$ to obtain the original distribution $x_0$, asymptotically. As depicted in Figure 2, the reverse process of each step can be represented as

$$p_\theta(x_{t-1}|x_t) = \mathcal{N}(x_{t-1}; \mu_\theta(x_t, t), \sigma_t^2 I), \quad (4)$$

where the $\mu_\theta(x_t, t)$ is predicted mean, $\theta$ is model parameter and $\sigma_t^2$ is associated with $\beta_t$. Similarly, in our work, we employ the Answer Denoiser to achieve this reverse process. However, we predict the distribution of candidate answers instead of $\mu_\theta(x_t, t)$, like [25].

## 4 METHOD

The main structure of DiffAns is illustrated in Figure 3, comprising three modules. Firstly, Encoders achieve the encoding of both video and question, effectively mapping the visual modality into the semantic space of text modality, and facilitating contextual understanding of question. Subsequently, the Answer-aware Condition Generator interact and fuse the video and question to obtain answer-aware condition, essential for guiding the subsequent denoising process. Lastly, Answer Denoiser iterate to denoise the noised probability distribution integrating noise intensity, which simulate the coars-to-fine filtering on vast candidate answers.

### 4.1 Video and Question Encoders

For the textual input (e.i., question $Q$), we introduce the robust RoBERTa[31] in NLP as our language backbone. Specifically, $Q$ is initially treated as a text sequence of length $l_t$. If the sequence length is insufficient, padding is applied; otherwise, the sequence is truncated. Subsequently, they are tokenized and passed through the pre-trained RoBERTa, where the output of the last hidden layer serves as the representation of question (e.i., $Q_{emb} \in R^{l_t \times d_t}$), where $d_t$ denotes the hidden size of RoBERTa. To mitigate potential biases introduced by the pre-trained RoBERTa, we employ Encoder in Transformer[42] (e.i., a multi-head attention layer (ATTN) followed by a feedforward network (FFN)) to perceive the contextual information of question. This can be defined as

$$Q_{enc} = LN(FNN(ATTN(Q_{emb}, Q_{emb}, Q_{emb}))), \quad (5)$$

where $LN$ denotes a regularization method (i.e. LayerNorm). The definition of ATTN is:

$$attn(Q, K, V) = Softmax(\frac{QK^T}{\sqrt{\tau}})V, \quad (6)$$

$$ATTN(Q, K, V) = attn(QW_1, KW_2, VW_3)W_4 + Q, \quad (7)$$

where $W_i \in R^{d \times d}$ are trainable parameters, and $d$ is the size of our model dimension, which is equal to $d_t$ in our implementation. $Softmax$ is an activation function, $\tau$ denotes the temperature of attention. The $attn(Q, K, V)$ represents scaled dot product attention with $Q$, $K$ and $V$ as the query, key and value respectively. And the definition of FNN is:

$$FNN(x) = ReLU(xW_1)W_2 + x, \quad (8)$$

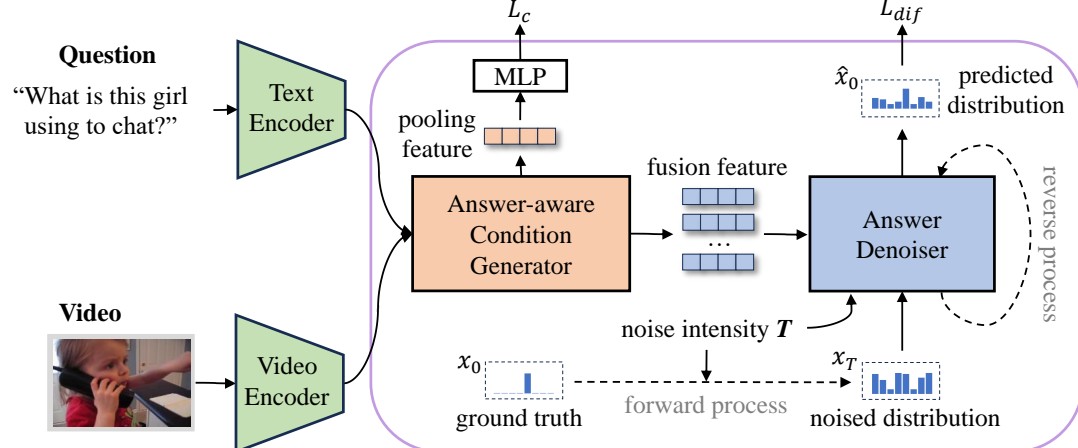

**Figure 3: The overall architecture of DiffAns, which contains three modules: (1) Encoders embed the video and question respectively, projecting multimodal features to consistent semantic space. (2) Answer-aware Condition Generator (ACG) interact and fuse the video and question, and obtains the condition that guides the following denoising process. (3) Answer Denoiser (AD) refines the noised answer distribution and predict the correct answer. Fusion feature is obtained by ACG and inputted into AD as condition. Pooling feature refers to the pooling of fusion feature in the temporal dimension. The dashed line represents the iterative process.**

where $W_i \in R^{d \times d}$ are trainable parameters, $ReLU$ represents an activation function.

For the visual input (e.i., video $\mathcal{V}$), we employ the ViT[9] as the visual backbone, which is a widely used and effective model in CV. Specifically, we first uniformly sample $l_v$ frames from the video. For video with fewer than $l_v$ frames, we duplicate the last frame to fill the gap. Subsequently, each frame is embedded by the ViT initialized by CLIP[37], resulting in video representation (e.i., $V_{emb} \in R^{l_v \times d_v}$). In order to align the visual and textual features in a unified semantic space, we froze the ViT and introduced projecter, a linear layer followed by LN. It project the visual feature into textual semantic space, which can be represented as follow:

$$V_{enc} = LN(V_{emb}W + b), \qquad (9)$$

where $W \in R^{d_v \times d}$ and $b \in R^d$ are trainable parameters.

## 4.2 Answer-aware Condition Generator(ACG)

We aspire to achieve a comprehensive interaction and fusion of video $V_{enc}$ and question $Q_{enc}$, incorporating fusion features as condition for subsequent denoising process. This condition should encompass information for answering question, guiding the denoiser to refine candidates probability distribution more rapidly and accurately from random distributions. A intuition solution is the Decoder in Transformer[42], which comprises a self-attention layer, a cross-attention layer and a FFN layer. Specifically, it first learns potential query semantic of $Q_{enc}$ through self-attention to facilitate subsequent capture of video related information, which can be denoted as:

$$Q_c = ATTN(Q_{enc}, Q_{enc}, Q_{enc}), \qquad (10)$$

where $ATTN(Q, K, V)$ denotes the attention layer. Then, the cross-attention is employed with $V_{enc}$ as key and value and $Q_c$ as query

respectively. It utilizes attention mechanism to focus on video information relevant to the question. By removing redundant information, the model obtains crucial feature for answering question. This process can be defined as

$$F_c = ATTN(Q_c, V_{enc}, V_{enc}). \qquad (11)$$

Finally, we further explore deep semantic information through a FFN layer followed by LN to abstract the fusion of video and question, obtaining a more refined and higher-order condition. This can be denoted as

$$C = LN(FNN(F_c)). \qquad (12)$$

Due to the truth that different videos and questions require different answers to be generated. The denoising process heavily relies on the quality of the generated condition (e.i., $C$). To further ensure the quality of $C$, we retain the cross-entropy loss to constrain the condition generation process. We perform mean pooling on the fused features $C$ to obtain the global information. Then, we map it to the dimension of the answer distribution through MLP. These can be defined as

$$x_c = ELU(pooling(C)W_1)W_2, \qquad (13)$$

where $W_1 \in R^{d \times d}$ and $W_2 \in R^{d \times N_a}$ are trainable parameters, $pooling$ denotes the mean pooling on token length dimension. $ELU$ denotes an activation function. Then, we employ cross-entropy loss to constrain the relationship between this distribution and the gold label, with the formula as

$$L_c = -\sum_{i=1}^{N_a} z_i ln(x_c^i), \qquad (14)$$

where $z_i = 1$ if $i$ is the index of ground-truth answer $a^*$ and 0 otherwise. $N_a$ denotes the size of candidate answer set.

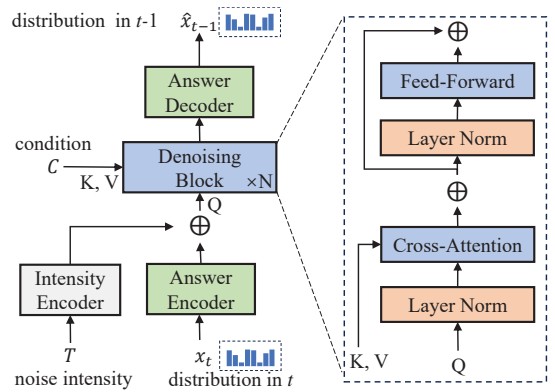

**Figure 4: The Answer Denoiser module, refining the noised answer distribution aggregating the intensity, under the guidence of answer-aware condition.**

Obviously, through these operations, the $C$ is expected to yield real answers $a^*$. Therefore, the condition $C$ inevitably possesses complete information for answer generation, bringing superior guidance to the subsequent denoiser.

### 4.3 Answer Denoiser(AD)

To mitigate the difficulty caused by selecting a single answer $a^*$ from a vast answer set $\mathcal{A}$, we introduce the conditional diffusion models, which progressively refine a random probability distribution $x_T$ to the correct distribution $x_0$ over multiple steps $T$ using the Answer Denoiser (AD), as shown in Figure 4. At each iteration, AD gradually refine the noised probability distribution, mimicking the direction selection and narrowing of the answer scope. Specifically, this process models $p_\theta(x_{t-1}|x_t)$ by mapping $x_t$ to denoised $x_{t-1}$. AD mainly comprises three components: Answer Encoder module maps the candidates probability distribution $x_t$ into the semantic space. Denoising Block module refines the feature of the noised distribution under guidance of condition $C$, and finally Answer Decoder module decodes the embedded distribution feature to obtain the denoised probability distribution $x_{t-1}$ after the current step.

For Answer Encoder, initially, during the forward process we need to continuously add noise into original answer distribution. Unlike generative tasks[41], the ground truth distribution is a one-hot representation in scope of $[0,1]$. Therefore, in order to align it with noise in the distribution, we first scale the distribution to the $[-\lambda, \lambda]$ to better fit the Gaussian distribution, following previous method[6, 25]. This is represented as

$$x_0 = \lambda(x_0 - 0.5).\tag{15}$$

After noise addition from $x_0$ to $x_T$, it is rescaled back to the original $[0,1]$ which can be denoted as

$$x_T = x_T/\lambda + 0.5.\tag{16}$$

We encode the distribution of noisy $x_t$ using a linear layer, mapping it to the semantic space of the model, denoted as

$$x_{enc} = x_t W + b,\tag{17}$$

where $W \in R^{N_a \times d}$ and $b \in R^d$ are trainable parameters.

For Denoising Block, we introduce information of noise intensity $T$ to aid in the denoising process. We encode $T$ using a sinusoidal mapping and add it into $x_{enc}$, obtaining the intensity-aware noisy answer encoding $x_{int}$. There we stack $N$ blocks based on cross-attention and FFN to achieve denoising of $x_{int}$ under guidance of condition $C$. In each block, we treat $x_{int}$ as query and $C$ as key and value, respectively. These are interacted via cross-attention, yielding an answer distribution $x_c$ fused with conditional information. This is formulated as

$$x_c = ATTN(x_{int}, C, C).\tag{18}$$

Then, we refine the answer distribution through a FNN layer, represented as

$$x_r = FFN(x_c).\tag{19}$$

In the above two formulations, we omitted LayerNorm for simplicity, which specific position in module can be seen in Figure 4. Undergoing $N$ blocks, we achieve the denoising process for the step $t$, obtaining the refined answer encoding for $x_r$.

For Answer Decoder, after a LayerNorm, we decode the encoded information $x_r$ with MLP, transforming $x_r$ into $x_{t-1}$. This process can be represented as

$$x_{t-1} = ELU(LN(x_r)W_1)W_2,\tag{20}$$

where $W_1 \in R^{d \times d}$ and $W_1 \in R^{d \times N_a}$ are trainable parameters. After multiple iterations of denoising over several time steps $T$, we derive the predicted $\hat{x}_0$. Then, we employ the Kullback-Leibler divergence as loss of $\hat{x}_0$ and $x_0$, aiming to minimize the distributional gap between them. This can be denoted as

$$L_{dif} = KL(x_0 \| \hat{x}_0).\tag{21}$$

### 4.4 Training and Inference

During the training process, the true label distribution $x_0$ is initially mapped to a distribution $x_T$ close to Gaussian noise through a forward process. Subsequently, Answer Denoiser is applied in the reverse process, effectively denoising $x_T$ under guidance condition $C$. After multiple iterations to obtaine the predicted probability distribution $\hat{x}_0$, the diffusion loss $L_{dif}$ is calculated. Additionally, to ensure the quality of the generated condition, $L_c$ is incorporated into Answer-aware Condition Generator. Therefore, the overall loss during training can be expressed as

$$L_{all} = L_{dif} + \alpha L_c,\tag{22}$$

where $\alpha$ is hyparameter to control the weight of $L_c$.

During the inference process, without ground truth answer, only the reverse process is feasible. The corresponding noisy distribution $x_T$ is replaced with randomly generated Gaussian noise. The model graduallty refines the random distribution through the reverse process and obtains the final predicted probability distribution $\hat{x}_0$. The ultimate prediction class label $\hat{a}$ is the candidate answer with the highest probability score, represented as follows:

$$\hat{a} = \arg\max(\hat{x}_0).\tag{23}$$

## 5 EXPERIMENTS

### 5.1 Experimental settings

*5.1.1 Datasets.* We evaluated our method with three widely used and challenging datasets in the open-ended VideoQA: MSVD-QA[48],

**Table 1: Statistics on datasets.**

| Dataset | #Video | #QA pair | $*\mathcal{V}$ | $*\mathcal{Q}$ | $|\mathcal{A}|$ |
|---|---|---|---|---|---|
| MSVD[48] | 2K | 50K | 10s | 6.6 | 2081 |
| MSRVTT[49] | 10K | 244K | 15s | 7.4 | 4000 |
| TGIF-Frame[18] | 39.5K | 53.1K | 3s | 9.8 | 1541 |

MSRVTT-QA[49] and TGIF-FrameQA[18]. The statistics on these three datasets can be seen in Table 1.

MSVD-QA[48] encompasses a plethora of topics, including humans, animals, various actions, and unique scenarios, providing rich content for evaluation. It comprises approximately 2k videos, with around 50k samples (i.e., QA pairs) in total. On average, each video has 25 related questions. The average video length is 10 seconds, and the average number of words in question is 6.6. The size of the candidate answer set is around 2k.

MSRVTT-QA[49] represents a broader compilation, showcasing a more diverse visual content, including the most comprehensive categories. It contains around 10k videos, with approximately 244k samples in total. Similar to MSVD-QA, each video has an average of 24.4 questions. The average video length is 15 seconds, the longest among the three datasets. The average question length is 7.4, longer compared to MSVD-QA. The size of the candidate answer set is 4k, which is the largest of these dataset.

TGIF-FrameQA[18] features videos sourced from real internet GIFs, accompanied by human-generated questions and answers, enhancing the authenticity. It comprises an astounding number of approximately 39.5k videos, with around 53.1k samples in total, implying that each video has fewer than 2 related questions on average. The average video length is 3 seconds due to the GIF format, and the average question length is 9.8, the longest among the three datasets. The size of the candidate answer set is 1,541.

For all three datasets, we followed previous works and evaluated the models using Accuracy(%) as the metric.

*5.1.2 Implementation Details.* In the feature representation module, we sample $l_v = 16$ frames for video and fix the maximum length of input sequences to $l_t = 20$ tokens. Subsequently, we employ ViT-L14 initialized by CLIP[37] and RoBERTa-base [31] as the visual and language backbones respectively, both with the hidden dimension size $d_v = d_t = 768$. We froze the ViT and fine-tune the RoBERTa. The model is configured with a same dimension size $d = 768$ of language backbone. For the MSVD-QA, MSRVTT-QA, and TGIF-FrameQA datasets, the numbers of denoising blocks $N$ are set to $\{4, 2, 3\}$ respectively, with a loss weight $\alpha = 1$. Within the diffusion model, we utilize DDIM[41] as the diffusion strategy, cosine schedule for noise, with the scale $\lambda = 0.5$. And the sampling steps (e.i. noise intensity) $T = 50$. During training, the model's learning rate is $1e - 4$, with a batch size of 128, and a maximum iteration of 10.

## 5.2 Main Result

We compared our DiffAns model with recent SoTA methods on three datasets in Table 2. To highlight the competitiveness of our method, we analyze the result from three perspectives.

*5.2.1 Answer Generation.* Most SoTA methods addressing open-ended VideoQA are classifier-based, exhibiting commendable effectiveness. However, some recent approaches such as CoVGT[47] and ATM[5], have also proposed the utilization of similarity-based method. Unlike other methods, they both have all embraced pre-training strategy. It is evident that similarity method is more suitable for models with pre-training process; otherwise, classifier method are sufficient to achieve commendable results, such as TranSTR[27] and KPI[24], which achieve the best performance. Given the large size of answer set in open-ended VideoQA, both of the aforementioned models inevitably introduce difficulty by selecting a unique answer from such a multitude of candidate answers in a single step. Therefore, we opt to utilize a diffusion-based method to refine the predicted answer gradually, marking the first employment of diffusion methods in VideoQA. Ultimately, our results surpass all recent SoTA methods, whether classifier-based or similarity-based. This demonstrat the effectiveness of our proposed DiffAns model.

*5.2.2 Backbone.* As time goes on, an increasing number of advanced backbones have been proposed and applied in the field of videoQA. In the early time, SoTA methods mostly employ CNN-based model [1, 4, 10, 13, 14] as visual backbone and utilize word embedding[36] to encode question. With the success and application of Transformer[42], it has also been introduced into VideoQA, achieving remarkable performance. Many methods have begun to use them as the backbone, whether in the field of visual[9] or language[8, 15, 31] domain. The two modal backbones utilized in our model are respectively the commonly used and efficient ViT[9] and RoBERTa[31]. Compared to our model, TranSTR[27] incorporates Faster R-CNN[1] for extracting object features in addition to ViT for the visual aspect. Moreover, a more advanced language model(e.i., Deberta[15]) is utilized as language backbone. However, their ultimate performance still falls short of our model's. This further substantiates the effectiveness of the proposed diffusion-based paradigm.

*5.2.3 Answer Set Size.* Meanwhile, our analysis reveals that the difficulty of these three datasets precisely corresponds to the size of the answer set (lower accuracy indicating higher difficulty). TGIF-FrameQA[18] has the smallest answer set, with the highest accuracy, while MSRVTT-QA[49] has the largest answer set, with the lowest accuracy. Thus, it can be concluded that the broader answer scope in open-ended VideoQA leads to increased interference in model selection of the correct answer from numerous candidates, rendering the dataset more challenging. The proposed DiffAns focuses on coarst-to-fine filtering candidate answers to address open-ended VideoQA. Experimental results demonstrate that our approach surpass SoTA methods, validating the effectiveness of multi-step denoising for selecting correct answer from large answer sets.

## 5.3 Modules Analysis

*5.3.1 Model Design.* We conducted ablation experiments on the three main components of our model: Encoders, Answer-aware Conditon Generator (ACG) and Answer Denoiser(AD). The results are presented in Table 3, where Proj and Enc denote the visual projecter and question encoder in Encoders, respectively. The model in the first row represents our baseline architecture, consisting of

**Table 2: Comparison to recent SoTA methods. Bold represents the highest, underline represents the second highest.**

| Method | Venus | Answer | Visual | Language | MSVD | MSRVTT | TGIF-Frame |
|---|---|---|---|---|---|---|---|
| B2A[32] | CVPR21 | Classifier | ResNet, ResNeXt | GloVe | 37.2 | 36.9 | 57.5 |
| HOSTR[7] | IJCAI21 | Classifier | ResNet, ResNeXt, Faster-R-CNN | GloVe | 39.4 | 35.9 | 58.2 |
| MSPAN[12] | ACL21 | Classifier | ResNet, ResNeXt | GloVe | 40.3 | 37.8 | 59.7 |
| MHN[34] | IJCAI22 | Classifier | ResNet, ResNeXt | Glove | 40.4 | 38.6 | 58.1 |
| HQGA[45] | AAAI22 | Classifier | ResNeXt, Faster-R-CNN | BERT | 41.2 | 38.6 | 61.3 |
| EIGV[26] | MM22 | Classifier | ResNet, ResNeXt | BERT | 42.6 | 39.3 | - |
| MMQEN[30] | MM23 | Classifier | ResNet, Faster-R-CNN, I3D | RoBERTa | 41.4 | 37.9 | 61.0 |
| PMT[35] | AAAI23 | Classifier | X3D | Glove | 41.8 | 40.3 | 60.6 |
| ATM[5] | MM23 | Similarity | ResNet, Faster-R-CNN | BERT | - | 40.3 | 61.6 |
| CoVGT[47] | TPAMI23 | Similarity | ResNet, Faster R-CNN | RoBERTa | - | 40.0 | 61.7 |
| KPI[24] | ICCV23 | Classifier | ResNet, ResNeXt, Faster-R-CNN | BERT | 43.3 | 40.0 | 63.0 |
| RaFormer[28] | MM23 | Classifier | ViT, Faster-RCNN | DeBERTa | 46.0 | 42.3 | - |
| TranSTR[27] | ICCV23 | Classifier | ViT, Faster-RCNN | DeBERTa | 47.1 | 43.1 | - |
| DiffAns | - | Diffusion | ViT | RoBERTa | **49.1** | **43.3** | **65.4** |

**Table 3: Analysis of model design. AD means Answer Denoiser, ACG means Answer-aware Condition Generator. Proj and Enc denote the visual projecter and question encoder in Encoders, respectively.**

| Model | MSVD | MSRVTT | TGIF-Frame |
|---|---|---|---|
| Linear | 46.8 | 42.4 | 64.5 |
| + AD | 49.0 | 42.6 | 64.7 |
| + ACG | 49.0 | 43.0 | 65.1 |
| + Proj&Enc | **49.1** | **43.3** | **65.4** |

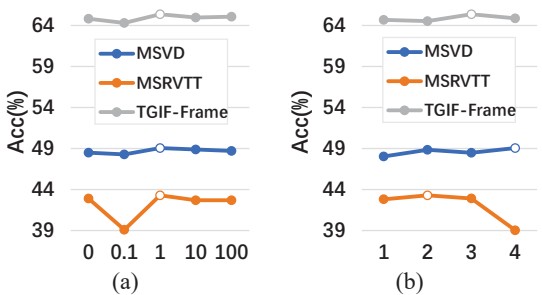

**Figure 5: Analysis of (a)$\alpha$ and (b)$N$. The white dots represent the highest accuracy.**

backbones followed by concatenating and a linear layer. Each subsequent row corresponds to the preceding model plus the new module. It demonstrates that with each module added, the performance of the model improves incrementally. Specifically, employing the diffusion model (e.i., AD) enhances the model's performance across all three datasets. Furthermore, leveraging ACG to refine learned condition in significant improvements for both MSRVTT-QA and TGIF-FrameQA. This underscores how answer-aware condition generated by ACG greatly aids the denoising process of AD. Subsequently, the incorporation of Proj and Enc further benefit the model, leading to better performance across all datasets.

*5.3.2 Loss.* The model is jointly trained with two losses (e.i., condition loss $L_c$ and diffusion loss $L_{dif}$). We analyze the influence of loss weight $\alpha$ on the model, which represents the importance of $L_c$, as shown in Figure 5(a). (1) Comparing with the boundary ($\alpha = 0$), which implies the model only utilizes $L_{dif}$ without $L_c$, the performance of the model is lower compared to joint training ($\alpha = 1$). This demonstrates the effectiveness of constraining ACG to achieve better guidance for AD. (2) When $\alpha$ is too low or too high, the model's accuracy decreases, especially at 0.1. This indicates that excessively low $\alpha$ not only lack the ability to learn high-quality condition, but also interfere with the learning of the main task (e.i., $L_{dif}$). On the other hand, when $\alpha$ is too high, the model focuses excessively on learning condition, thereby losing denoising ability.

Finally, an appropriate $\alpha$ can make the two losses more compatible, leading to optimal result.

*5.3.3 Denoising Block.* We explore the impact of $N$, which denotes the number of denoising blocks, as shown in Table 5(b). It proves that too large or too small $N$ leads to decrease of model performance. This phenomenon may arise from insufficient layers causing inadequate denoising and excessive layers leading to overfitting.

## 5.4 Diffusion Analysis

*5.4.1 Impact of Design.* We analyzed the usage of noise intensity $T$, $L_{dif}$ type, diffusion strategy and the schedule of $\beta$ in diffusion model, as shown in Table 4a. (1) After removing the $T$ when denoising, the model experiences significant declines on MSRVTT-QA and TGIF-FrameQA. It proves that the perception of noise intensity can improve denoiser. (2) We utilized another commonly used loss MSE as $L_{dif}$. We find that the effectiveness of KL divergence is superior, possibly due to its better suitability for measuring the similarity between two distributions. (3) DDPM[17] and DDIM[41] are two commonly used strategies for diffusion. DDPM learns the latent data distribution from Markov chain, while DDIM models a non-Markovian process, accelerating the sampling process. As

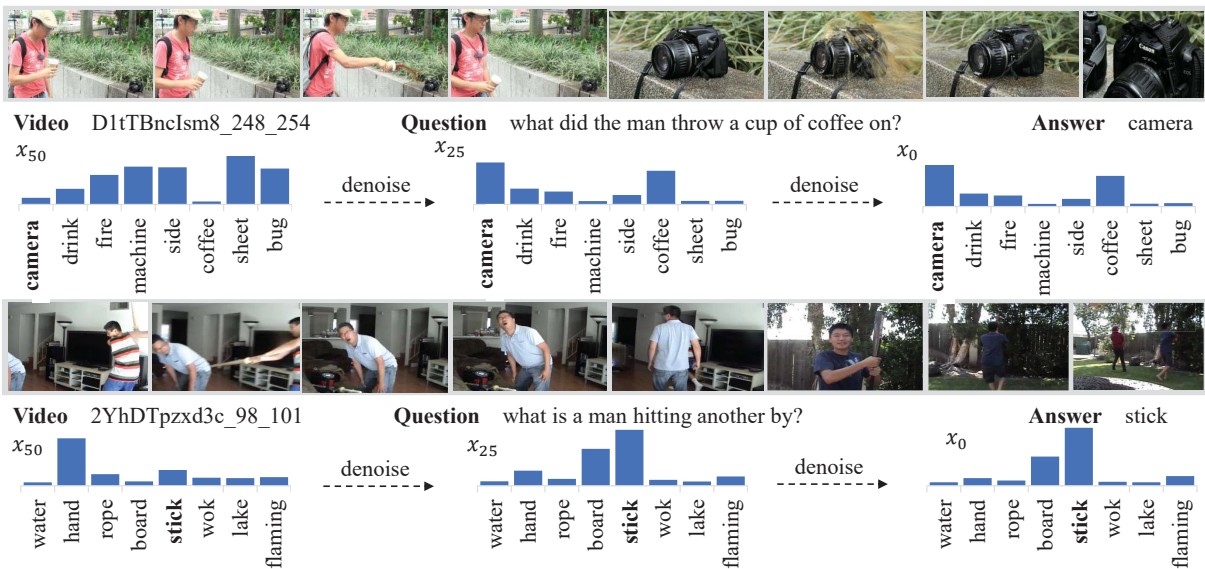

Figure 6: The inference process of DiffAns, demonstrating the answer distributions at different number of denoising steps.

**Table 4: Diffusion Analysis.**

**(a) Variants of diffusion model.**

| variant | MSVD | MSRVTT | TGIF-Frame |
|---|---|---|---|
| w/o intensity $T$ | 49.2 | 42.7 | 64.9 |
| KL → MSE | 19.3 | 31.8 | 35.0 |
| DDIM → DDPM | 48.7 | 40.8 | 64.6 |
| cosine → linear | 48.4 | 33.0 | 62.0 |
| DiffAns | **49.1** | **43.3** | **65.4** |

**(b) Impact of scale $\lambda$.**

| scale $\lambda$ | MSVD | MSRVTT | TGIF-Frame |
|---|---|---|---|
| 0.2 | 49.0 | 43.0 | 65.4 |
| 0.5 | **49.1** | **43.3** | **65.4** |
| 1 | 48.9 | 42.7 | 64.7 |
| 2 | 48.9 | 41.0 | 64.4 |

**(c) Impact of steps $T$. The t denotes the average time of inferring one sample on TGIF-Frame.**

| steps $T$ | MSVD | MSRVTT | TGIF-Frame | t |
|---|---|---|---|---|
| 10 | **49.2** | 42.5 | 65.2 | 0.15ms |
| 50 | 49.1 | **43.3** | **65.4** | 0.36ms |
| 100 | 48.6 | 42.8 | 65.3 | 0.53ms |
| 1000 | 48.8 | 42.6 | 65.4 | 4.71ms |

indicated, DDIM still outperforms DDPM in performance, thus we adopted DDIM. (4) The schedule of $\beta$ in diffusion model determines how the step size increases. When replacing cosine with linear, the decrease of model performance is observed, particularly on MSRVTT-QA. Therefore, cosine schedule is more suitable and is utilized in our model.

*5.4.2 Impact of scale $\lambda$.* To better align the probability distribution with Gaussian distribution for enhanced noise integration, we applied scaling process. The results under different $\lambda$ are presented in Table 4b. It is observed that the model achieved optimal performance when $\lambda = 0.5$, indicating its suitability. Our analysis suggests that excessively low $\lambda$ lead to information loss, while overly high $\lambda$ hinder the model to develop robust denoising capabilities.

*5.4.3 Impact of steps $T$.* We investigated the impact of varying $T$ on the model's performance, as depicted in Table 4c, where t denotes the average time of inferring one sample on TGIF-FrameQA. A higher $T$ incurred greater computational costs. It was observed that our model achieve optimal performance when $T = 50$. This contrasts with tasks such as Image Generation, where 1000 or more steps are commonly utilized. This difference can be attributed to the requirement of precise pixel-level generation in image tasks[41], whereas our model focuses on deriving a probability distribution.

## 5.5 Visualization

DiffAns differs from other conventional methods in that model iteratively denoising the answers through multi-step refinement, gradually shaping the correct probability distribution. Consequently, we visualize the distributions at different step of inference on MSVD-QA dataset, as depicted in Figure 6. Due to space constraints, we illustrate the probability of top 8 candidate answers, which is refined from random to biased during inference. The phenomenon meet expectation that firstly the answers relevant to video or question get top probability, but finally ground truth is higher while others decrease. In that practice, we also find that less step is enough, which reason may be same to 5.4.3.

## 6 CONCLUSION

Open-ended VideoQA involve a vast candidate answer set. Selecting a single answer in a one-step manner without coarse-to-fine filtering leads to inaccuracy. Hence, we introduce a diffusion model that gradually refines answers through multi-step denoising, achieving a more precise answer generation process. This marks the first application of diffusion model to VideoQA, and our experiments corroborate the efficacy of our approach. We aspire that our innovative work will bring insights to open-ended VideoQA field.

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
