# OpenReview forum: "Conditional Diffusion Model for Open-ended Video Question Answering"
_acmmm.org/ACMMM/2024/Conference — MM2024 Poster_

### Official Review · Reviewer_nap2 · 2024-05-22

**Rating:** 4
**Confidence:** 3

**Summary:**

The paper proposes a diffusion-based method to apply the Open-ended VideoQA task by simulating the gradual process by which humans answer open-ended questions. The difference between the method compared with traditional methods such as the classifier method and similarity method is the model iteratively denoises to obtain the correct probability distribution. In the train procession, the true answer labels are fusion in the video and questions encoder to help better get the final correct answer label. The method was implemented in the three open-ended VideoQA datasets. The extensive experiment results verify the superiority of this method.

Questions
1.How does the answer denoiser reflect the coarse-to-fine filtering on vast candidate answers? I don't seem to see this process of "filtering".
2.In the main experimental results presented in Table 2, it is observed that for the largest answer set scale dataset, MSRVTT, the proposed method did not show significant improvement compared to the classifier method TranSTR. This makes it difficult to convincingly demonstrate the capability of the proposed method in handling datasets with large answer sets. It is recommended to provide comparisons with more similar datasets to further substantiate these findings.

**Strengths:**

1.The approach innovatively proposes solving the open-ended VideoQA task using the diffusion-based model, which has better interpretability than traditional approaches.
2.This paper demonstrates a well-structured framework, with clear organization and logical flow throughout the content.

**Limitations:**

1.In the experimental section, the dataset used is too limited. It would be more persuasive if more datasets were utilized.
2.In the comparative experiments, the classifier method was extensively utilized, but there was insufficient comparison for the similarity method.
3.It fails to delve deeper into the reasons that would better illustrate the effectiveness of this method.

**Suitability:**

2

---

### Official Review · Reviewer_mHGK · 2024-05-24

**Rating:** 3
**Confidence:** 3

**Summary:**

The paper introduces a novel model named DiffAns, which employs a conditional diffusion model to address the challenge of open-ended Video Question Answering. Unlike multiple-choice VideoQA, open-ended VideoQA requires identifying the correct answer from a vast pool of potential answers without any fixed options. The DiffAns model simulates the human process of gradually refining the probability distribution of candidate answers to achieve more precise predictions. It integrates a diffusion model that first diffuses the true answer into a random distribution and then iteratively denoises it to retrieve the correct probability distribution under the guidance of an answer-aware condition.

**Strengths:**

(1) The DiffAns model seems to pioneer the use of diffusion models in the field of VideoQA, providing a new perspective on how to handle the open-ended nature of the task.

(2) The paper's experimental results appear promising, indicating potential for further exploration and application in the field.

**Limitations:**

(1) The motivation of this paper is unclear. The authors claim that gradual refining of the predicted distribution can achieve a more precise prediction for VQA since this process simulates human intuition. However, this explanation lacks sufficient (theoretical and/or empirical) evidence. The authors only show the denoising process at very limited steps (t=50, 25, and 0) in Figure 6, which cannot directly illustrate why and how the proposed method works well (by simulating human intuition). The lack of explanation of motivation and method makes it difficult to judge the scope and applicability of the proposed method.

(2) The English writing of this article leaves much to be desired. I think the organization and logic of the article are clear. However, there are too many grammatical mistakes. For example, “seek” should be “seeks” in line 40. “use” should be “uses” in line 95. “search” should be “searches” in line 115. “encode” should be “encodes” in line 149. “simulate” should be “simulates” in line 220. “e.i.” should be “i.e.” in lines 320, 326 and 329. “demonstrate” should be “demonstrates” in line 655. I encourage the authors to polish their writing carefully.

(3) Since DeBERTa is a more advanced language model than Roberta (as stated in line 669), why not use it as a language backbone in this paper?

(4) Why are the results of RaFormer and TranSTR, two current SOTA methods, on TGIF-Frame missing?

**Suitability:**

3

---

### Official Review · Reviewer_goh6 · 2024-05-24

**Rating:** 4
**Confidence:** 3

**Summary:**

This paper proposes DiffAns as the first diffusion-based method in open-ended video question answering. In the forward pass, the true answer label is diffused into a random distribution. Then the model iteratively denoises to obtain the correct probability distribution under the guidance of answer-aware condition generated from video-question information. Experiments are conducted to validate the proposed method.

**Strengths:**

1.The motivation of the paper is clear. Employing a diffusion model to construct the multi-step reasoning process of open-ended VideoQA is intuitively correct and reasonable.

2.This work introduces diffusion models into the domain of open-ended video question answering, which makes a relatively large technical contribution. Combining diffusion models with generative multimodal models such as VideoQA models can be an interesting and meaningful future research area that deserves further exploring.

3.The paper is well written and easy to follow.

**Limitations:**

1.Although integrating diffusion models with downstream VideoQA models is of value, exploring the combination of LLMs/LVLMs with the proposed diffusion-based framework can be more interesting and more promising.

2.This work verifies the idea and effectiveness of their designs on three datasets. I suggest more datasets can be included to further enhance the quality of the paper.

**Suitability:**

3

---

### Meta-Review · Area_Chair_M8AP · 2024-07-01

**Recommendation:** Accept (Poster)
**Confidence:** 5

**Metareview:**

This paper presents a conditional diffusion model for open-ended video question answering. It receives two borderline accept and one borderline reject. The response well addresses the reviewers' concerns. All reviewers give the borderline accept ratings after the rebuttal. The merits, including motivation, interesting ideas, good writing, and good results, are well recognized by the reviewers. Please also incorporate the analysis in response into the camera-ready version. Overall, I think the current manuscript meets the requirements of this top conference.